## Perspective

peer support group; experiential narrative; challenges and recommendations

**Corresponding author:**
Sneha Agarwal;
Email: snehaxyz02@gmail.com

# Peer support groups through an experiential lens

Sneha Agarwal[1] 🔘, Udisha Maurya[1] and Himanshu Kulkarni[1]

[1]Manorathi Foundation, Delhi, India

## Abstract

Peer support groups are safe spaces that provide an emotionally and socially supportive environment to individuals, along with practical assistance. The three authors run a peer support group in New Delhi, India, for young adults who face mental health challenges. This study explores the processes of group formation, the nature of participant engagement and the evolving dynamics within the group setting. Drawing on firsthand reflections, the article highlights how peer support groups foster emotional safety, narrative autonomy and identity reconstruction through shared lived experiences. It also outlines the practical challenges of facilitation, including managing boundaries, maintaining group cohesion and adapting to diverse participant needs. The article concludes with mentioning arenas of further growth.

## Impact statement

This experiential narrative seeks to illuminate the emotional, relational and practical dimensions of peer support that are often overlooked in conventional academic or clinical discourse. By centering lived experience, this study affirms it as a legitimate and powerful form of knowledge. Through practice-based reflections and illustrative vignettes, it offers a grounded portrayal of peer support groups as spaces where individuals are seen, heard and understood without judgment. These accounts highlight the personal growth of both participants and facilitators, reinforcing the value of peer support as an effective and meaningful social support system.

In doing so, this study aims to inspire and guide facilitators, mental health practitioners and youth workers who are working to build inclusive and empowering spaces that prioritize relational healing and shared agency. It also invites policymakers to consider peer support groups as a viable, community-rooted approach to youth mental health, particularly in low- and middle-income contexts like India, where access to formal care may be limited. Ultimately, this study contributes to the growing body of literature that recognizes the transformative potential of peer support within the field of community mental health.

## Introduction

*NS is an 18-year-old queer person, navigating a complex history of trauma and abuse. Given the stigma prevalent in his sociocultural context, he has no place to call his own. He does not find belonging even in the queer community. He is reluctant to start therapy, stemming from a fear of further marginalization.*

*AB is a 20-year-old boy who was diagnosed with early-onset schizophrenia at the age of 15 years. Having spent extensive time in hospitals, followed by therapy, he is considered 'recovered'. However, he still has moderate support needs, a strong urge to tell his story and to help others in return.*

These two stories – distinct in their trajectories yet unified by shared themes of isolation, recovery and the need for authentic connection – underscore the critical role of peer support in contemporary mental health care. Peer support is broadly defined as the emotional, social and practical assistance provided by individuals who share lived experiences of mental health challenges (Solomon, 2004). It is grounded in mutual understanding, shared responsibility and the principle of equality, offering nonhierarchical relationships that contrast sharply with the clinician–patient dynamic (Mead et al., 2001).

Peer support can occur in both informal community settings and formalized healthcare systems, contributing significantly to prevention, empowerment and recovery (World Health Organization, 2017). The value of peer support lies in its capacity to reduce isolation, counter stigma and build communities of care. Internationally, peer-led interventions have demonstrated significant improvements in self-efficacy, psychiatric symptom reduction, treatment adherence and community participation (Davidson et al., 2006; Gillard et al., 2014).

Similar outcomes are now being reported within India. The You're Wonderful Project in Delhi facilitates peer-led support groups – both online and offline – for young individuals facing

depression and anxiety, fostering openness and empathy. Quality-Rights Gujarat, supported by the World Health Organization, incorporates peer support into mental health institutions to safeguard patient rights and challenge coercive practices (Pathare et al., 2018). The Baatcheet digital platform, launched during the coronavirus 2019 pandemic, offers structured peer spaces for emotional expression, especially among youth and frontline workers (Gonsalves, 2024).

There also exist instances of peer support in some schools in Mussoorie and Delhi. In a school located near a rag picker's colony, peer support is used along with games and other activities to integrate the children into school. Other examples of peer support include 'circle time' where children bring up topics of importance to them, like bullying, social media and time management. Buddy systems, where children with different interests, those new to the school or those feeling homesick are paired with other kids, is another example of peer support (Mathew, 2024).

To add to this growing body of literature, we have penned down our experiences of running a peer support group. By drawing from firsthand experience, we aim to illuminate the emotional, relational and practical dimensions of peer support that often remain invisible in conventional academic or clinical writing. The intent is to reflect on what was learned, offer insight into lived experiences and guide others seeking to create safe, inclusive and empowering mental health spaces. In doing so, the study affirms the value of lived experience as a legitimate and powerful source of knowledge.

## The peer support group

The three authors are facilitators of a support group located in New Delhi, India. To explain what a support group means, we use *AB*'s (one of our participants) amazing analogy. He had reflected and said that we were sitting in different places in the same room, and we all viewed the room from different perspectives. The peer support group was a place to share what we perceive and hear what others see in the room.

To contextualize, the 'room' in our case refers to the common circumstances we share, that is, being a young adult in India, who has struggled or is struggling with mental health issues (diagnosed or undiagnosed). Both the participants (aged 18–25 years, both male and female, belonging to the urban middle class) and facilitators (aged 22–25 years, one male and two female, belonging to the urban middle class), referred to as peers, share these characteristics. Given the developmental age, points of struggle include managing stress, establishing an identity, career issues and relationship problems. Considering the family dynamics within the Indian context, balancing parental expectations with a need for independence also becomes a source of concern.

The group meets once a month for a time period of 2 h. It is an open group, that is, people are allowed to drop in and out. At any given point, we have 10–15 members, with the duration of membership varying from 3 months to 2 years. Having an option allows participants the right to choose whether they wish to be associated with the group or not, as well as allowing new members who may need support to join and bring in fresh perspectives (World Health Organization, 2017). It also brings forth challenges of continuity and stability, for which certain protocols have been set in place that are discussed later in the article.

The group is associated with a licensed therapist who has considerable experience in running support groups. She is our supervisor, and before starting the group, the authors were sensitized and trained by her. The authors were taught about setting ground rules and boundaries, how to practice active listening and respond with empathy, along with potential problems that could arise and how to deal with them (Community Tool Box, 2014). We were provided with reading materials, and we role-played certain techniques (like asking open questions and reflecting feelings) as well.

## Running the support group

After completing the training, participants were recruited. They primarily consisted of our supervisor's clients, who she felt needed social support instead of, or in addition to, therapy. Other sources of getting participants include social media posts and word-of-mouth information spread by the facilitators. Once participants were recruited, they were individually contacted beforehand to understand their reasons for joining the support group and to provide them with the space to share any additional information that we should be mindful of.

A group was formed, and a time and place were decided for the meeting. After the participants assembled, we started off with introductions, followed by clearly explaining what the purpose of a support group was and deciding on basic things like how often, when and where to meet. Ground rules were set, which consisted of confidentiality (not to disclose what is being discussed), basic etiquette (not interrupting others and being respectful and empathetic) and steps to be taken if someone feels uncomfortable or overwhelmed (like letting the facilitator know privately or stepping out of the room) (Family Support Network, 2010).

The initial few sessions were structured, wherein the topic was predecided and activities were planned. Topics like friendship or career were picked up and explored. Core values, such as mutual respect, affirmation and acceptance, were emphasized. Active facilitation was demonstrated, where we modeled active listening, providing support by validating a person's experiences, followed by others sharing their experiences, post which we collectively problem-solved.

The latter sessions were more unstructured, where we let participants decide on-spot what they wanted to talk about or pick up a topic from something someone said. The sessions still followed a basic structure, that is, beginning with a feelings check, asking how everybody has been since the last time we met and adhering to a time limit of 2 h. Otherwise, conversations were more free-flowing, and facilitators did not need to intervene much.

After the meeting was over, we would talk to our supervisor about what went right and what could have been done better. Post-session conversations included how to deal with participants depicting boundary issues, when and how to intervene in certain circumstances, contacting participants who may need additional support (psychiatric or psychological) and providing them referrals or discussing which conversations may have been better suited for other occasions (Community Tool Box, 2014).

## From the facilitators' lens

Being a facilitator is a challenging job as it comes with multiple responsibilities: maintaining group security, creating an emotionally safe climate, intervening in case of boundary issues, and so forth (Dillon and Hornstein, 2013). At the same time, it is a deeply rewarding experience as well. It has given us a sense of purpose, along with the satisfaction that we have created something meaningful that has an impact on people's lives.

Being a facilitator is akin to a balancing act between taking charge and taking a backseat. Sometimes, when everybody is hesitant to start, we need to begin with our reflections and engage people to set the tone for the group and encourage others to share. At other times, we need to put our needs on the back and do what the group requires or the situation demands (in such scenarios, it is advisable to discuss your needs with your supervisor later).

Having multiple facilitators is beneficial as it allows us the freedom to move fluidly between participant and facilitator roles. Being in a participant role has led to deeper group engagement, as it taught us how much courage it takes, not only in opening up and being vulnerable but also in responding to others' vulnerability. Taking a participant role has also helped dismantle hierarchies and foster a more egalitarian group environment, as participants see us not as authority figures but as one of them.

Self-reflexivity has been central to our growth as facilitators (Dillon and Hornstein, 2013). In the beginning, one of the authors tended to control and micromanage the group, and had doubts whether she would be able to connect to different people – those of the queer community, from lower socioeconomic strata or having borderline traits. Engaging in self-reflexive practice helped her realize her anxiety and assumptions. Seeking guidance from her supervisor and connecting with participants over shared experiences helped her shift her approach and challenge her biases.

Drawing boundaries around empathy is another skill that has to be consciously developed. Initially, one of the authors would get emotionally invested in a member's narrative and lose sight of the group's broader needs. With experience, she realized that as a facilitator, her empathy needs to have pragmatic bounds. While every feeling must be acknowledged, the depth of engagement must be guided by the space and time available. This realization has helped her support individual participants without compromising the group's shared well-being.

Emotional regulation, especially during stressful situations, is another core skill (Foye et al. 2025). On one occasion, one of the authors became overwhelmed because a participant crossed boundaries repeatedly. He reacted to their every word and constantly tried to do damage control. Being genuine and disclosing how something impacts you is important, but as a facilitator, you cannot let your emotions overpower you. So he went out of the room, took deep breaths and came back with greater clarity. He reoriented himself to where the group was and how best to move ahead.

### Reflecting on the participants' experiences

The therapeutic potential of peer support groups lies in their ability to foster connections among individuals sharing similar experiences, thereby reducing social isolation and loneliness (Gillard et al., 2017; Worrall et al., 2018). A case in point is *NS* (a queer individual), who often reported feeling isolated. However, upon joining the peer support group, he encountered others navigating similar challenges, including academic struggles, familial conflicts and interpersonal difficulties. By recognizing shared experiences and empathizing with others, *NS* transitioned from feelings of isolation and hurt to a sense of belonging and hope.

Peer support groups offer a safe space where individuals are accepted and respected (Ussher et al., 2006; Gillard et al., 2017). Many of our participants mention that they do not feel understood either by their parents or peers. There is always a fear of judgment

surrounding their interactions. However, since effort to create a safe space has been put in, by setting ground rules and expectations, along with the requisite facilitation, participants find themselves being vulnerable. A touching instance involved *PV*, a rather reserved individual, who could finally open up about her dying friend, her search for a lucky four-leaf clover to save her and how much it hurts to lose someone so close to you.

This scenario was also a beautiful moment depicting how support works. Nobody tried to patronize or silverline *PV's* situation or give unwanted advice. What support looked like was: giving hope (maybe things will become better), providing perspective (things she could do), nonverbal gestures (big hugs) and silent actions (helping her find a clover). It is such moments that cement support groups as safe spaces and provide others the courage to open up.

While emphasis is put on talking about one's experiences, it is important to assure people that their presence is valued and appreciated, and belonging to a support group is not conditional on their ability to contribute verbally. Being quiet provides people time to observe, gauge their position and reflect on what they wish to share (Dillon and Hornstein, 2013). A perfect example of this is *KP*, a very withdrawn individual, who did not feel the need to come out of his shell. However, by observing people in the group, he realized that he did want to share his experiences and feel connected to people.

One way to remove emphasis from speaking as the only way of opening up and providing much-needed change from routine groups is to hold activity-oriented sessions. We achieved this by holding a few informal art-based sessions (World Health Organization, 2017). Participants were provided materials like drawing books, color pencils, paint and clay, and were asked to draw and express whatever they felt like. *PV*, who is mostly rather quiet, drew a beautiful art piece which was much appreciated by all. She later confessed that it made her day as she felt capable and worthwhile after a long time.

Other activities, including participants singing or reading the poems and couplets they have written, are also conducted at the end of a session. This helps end the session on a light note, allowing people to express themselves and increasing bonding. It also allows people to create an identity outside of their problems. For instance, *AB*, a participant with schizophrenia, is not defined solely by his diagnosis. He often bakes for the group and shares couplets written by him. This allows him to develop an identity beyond a person with schizophrenia, be it that of a baker, poet or student.

Support groups allow narrative autonomy, which gives individuals a chance to regain agency and control over their lives (Ussher et al., 2006). However, this ability presents its own set of challenges. *DC*, a participant with borderline traits, had frequent fights with his mother and weaved a tale wherein he was always the one wronged. It was clear that he struggled with perspective-taking. Group members navigated this delicate situation by validating his emotions, offering alternative perspectives and facilitating a reality-checking discussion. This collective problem-solving exercise empowered *DC* to consider multiple viewpoints and develop coping strategies.

### Issues to consider

Facilitators bear responsibility for ensuring participant and group safety. They must prioritize participants' emotional well-being, ensuring the exchange during the meetings does not exacerbate

distress. During a particularly intense and emotion-heavy session, the group members, especially DC, became overwhelmed. We recognized our mistake in not addressing the distress promptly and sought guidance from our supervisor, who intervened by conducting a feelings check and validating participants' emotions.

Support groups are spaces where each person is acknowledged as an 'expert by experience', that is, hierarchies are diminished or broken (Dillon and Hornstein, 2013). It is important to reiterate to the participants that facilitators are not experts; their job is to simply ensure the smooth functioning of the group. To equalize power dynamics, the three authors followed a rotation mechanism, ensuring a different person facilitated each group. We were also careful not to take sides in case of disagreements among participants, simply acknowledging the validity of different perspectives (World Health Organization, 2017).

A major issue that crops up time and again is how some participants tend to take up more space while others remain passive. It is a dicey situation, since we do not want to intervene too much and control the natural group dynamics. While sometimes it is okay for a participant to take up more space, if it is a recurrent pattern, then it becomes important to gently interrupt the participant and try to involve others in the discussion (World Health Organization, 2017). It can usually be achieved by asking the participant if they would like to hear other people's perspectives and asking other participants what they think.

Among all this, it is also important to remember that peer support groups are not a 'one-size-fits-all' model. Not everyone benefits from peer support groups. Some of the attributes that need to be taken into account are emotional regulation skills, self-awareness and the ability to consider diverse perspectives of the participants. *VK*, a participant with cognitive deficits, could not successfully participate in the group meetings due to difficulty articulating thoughts and an inability to comprehend others' perspectives.

Finally, since ours is an open group, participant and facilitator turnover can affect group dynamics as well. To address this challenge, we implemented specific protocols. Dedicated sessions for departing members ensured closure, processing of emotions and acknowledgment of contributions. Training of existing participants as co-facilitators ensured group continuity and fostered a sense of shared responsibility. Facilitation helps participants gain confidence and claim the group space as their own. It also provides them with a sense of agency, enhances their voice and builds leadership skills.

## Conclusion

The experiential account of a peer support group run in New Delhi, India, underscores support groups as nonjudgmental spaces with an empowering environment that fosters agency and connection. It delves into the authors' experiences as facilitators, consisting of forming the support group, receiving requisite training and learning to deal with challenges. It also dives into participant experiences, highlighting their growth over a period of time, as they form strong connections and become each other's support systems. Finally, it also examines possible problems that arise while running a support group and details ways of tackling them.

Going forward, the authors hope to facilitate deeper conversations, particularly regarding issues like gender expectations, dealing with feelings of inadequacy and learning self-acceptance. They hope to train participants as facilitators, which would help take the group ahead and lead to personal development of the participants. Other plans include expanding the support group, working on inclusivity of diverse individuals and scaling the group in a virtual setup. Building a wider network of peer support groups would also allow for mutual learning and resource-sharing.

**Open peer review.** To view the open peer review materials for this article, please visit http://doi.org/10.1017/gmh.2025.10061.

**Acknowledgments.** We are grateful for the guidance, encouragement and feedback of Mrs. Mona Sharma Rana, without whose support this would not have been possible. We are also grateful for the contribution of the support group members who were kind enough to share their experiences with us.

**Author contribution.** Conceptualization: S.A. Writing – original draft: S.A., U.M. and H.K. Writing – review and editing: S.A. All authors approved the final submitted draft.

**Competing interests.** The authors declare none.

**Ethical standard.** The research meets all ethical guidelines, including adherence to the legal requirements of the study country.

**Note.** All participants' initials have been randomized, and no identifiable information has been used, thereby maintaining their anonymity.

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
