## [Reviewer Report]

Many thanks for the invitation to review this perspectives article. The authors advance a perspective in support of the integration of peer support group models of mental health care. This is initially supported on the premise that peer support groups are scalable and resource efficient, thereby increasing accessibility of affordable mental health care. In the body of the text, the authors draw upon brief case vignettes to explore the interpersonal and relational dynamics that a peer support model offers that extend beyond the dominant biomedical model of health to support psychosocial and mental wellbeing, as well as identifying the limits to this model. They conclude with recommendations drawn from a wide variety of task-sharing mental health care programs.

Whilst the points raised are interesting, there are areas where the manuscript would benefit from further clarity in its positioning and supporting literature. These considerations include:

The manuscript itself is only sparsely referenced, and doesn’t engage effectively with the vast literature on this topic, including from the Indian setting. This engagement is required to support the arguments being made - there are many claims to evidence of effectiveness, scalability, cost effectiveness etc all of which require substantiating. Please note: on p.2, column 2, line 42 there is (reference) which requires completing.

It would also help for the authors to more clearly position the piece - it felt like the aim was to focus on the perspective of facilitators of peer support groups on this topic, and that the authors have this experience? If so, it would be useful to discuss this positionality of the authors to situate the contributions being made - i.e. that these are practice-based reflections. This would be a welcome contribution to the Global Mental Health literature. This would also orient away from arguments of scalability and resource efficiency, which are not well advanced in the manuscript. Instead, the focus would orient to the experiences of facilitating such groups.

In this positioning, it would also help to clearly define what it meant by ‘peer support group’. Are these groups facilitated by persons with lived-experience? Or does ‘peer’ refer to other characteristics? What training and ongoing supervision is provided to facilitators? How are groups formed? What do they provide support with? Do they have a structure or are they unstructured? How are the groups networked with other sources of support, including for ongoing referrals? Some of these points come through in the recommendations, but these seem to be drawn from wider global task-sharing initiatives rather than grounded in the groups being discussed in the vingettes, which weakens their relevance to the arguments being made.

I hope this feedback is supportive in more clearly conceptualising how this manuscript is positioned and the contributions it is making to the existing GMH literature.

---

## [Reviewer Report]

The authors put forth a Perspective on peer support groups in India. I think this piece considers an important topic—how do we scale and sustain peer support as a means of addressing the global mental health treatment gap. However, I think it could benefit from major revisions to make a strong contribution to the literature. In general, I think more references could be helpful throughout. Additionally, I think more clarity on what the authors are aiming to accomplish with this Perspective could be helpful. For example, are they aiming to describe their experiences? If so, more information on those experiences would be helpful. Or, as is mentioned in the Impact Statement, are they aiming to facilitate a paradigm shift? If so, more information on what that might look like is needed.

Abstract and Impact Statement:

I think the Abstract is well-written; however, the “Impact Statement” seemed a bit disconnected from the paper. First, I would suggest the authors edit the first paragraph of the Impact Statement and ensure that it’s clear that you’re discussing mental health problems and peer support for mental health problems. Then, in the latter paragraph, I would ensure that all claims are in line with what you offer in the manuscript. For example, only in the Impact Statement do the authors mention that these insights are based off “the implementation of a peer based support program for young people in the urban area of the capital city of New Delhi in India.”

Introduction:

I would suggest that the authors make their introduction paragraphs more specific to mental health and incorporate citations to support their claims. Currently, the claims are related to the broader healthcare system and lack citation.

More is needed on peer support groups themselves. What do the authors mean when they use this term? What do peer support groups look like in practice? What is the evidence for their effectiveness?

I also think that the introduction would benefit from a paragraph describing the context in which this group works. That is, explaining the “peer based support program for young people in the urban area of the capital city of New Delhi in India” that is referenced in the Impact Statement would help establish this group’s credibility and provide important context for the reader. This may also fit into the above point about better describing peer support groups.

From a facilitator’s lens:

This section seems to describe participants experiences, so as a reader, I felt like the heading “from a facilitator’s lens” did not seem quite accurate.

Things to Consider:

More operational guidance may be helpful in this section. What would your group concretely offer to others considering starting a peer support program? For example, you mention “not everyone benefits from peer support groups.” How do you select folks who are likely to benefit and/or monitor if folks are benefitting?

Peer Support in India:

This section may be better suited earlier in the manuscript. I believe describing it after describing your model of peer support might flow better, as right now it seems to come a bit too late after you’ve discussed implementation considerations.

A reference is missing after you describe the the QualityRights project in Gujarat.

---

## [Editor Report]

May you kindly address the reviewers' comments, particularly providing a granulated description of the intervention, including providing references throughout the scripture. Importantly, the introduction should be more streamlined for global mental health.

---

## [Reviewer Report]

Many thanks to the authors for their constructive engagement with initial review comments. I enjoyed reading this revised perspective piece, and felt the adoption of an authoritative voice of the lived-experience perspectives of facilitators to be an effective re-orientation of this manuscript which adds to the current literature on global mental health task-sharing and peer support models. Whilst I remain supportive of the publication of this piece, I would like to recommend further minor revisions:

Further clarification of the positionality of the three authors. For example, gender identities, ages / age range, socioeconomic background, what you are students of, and discussion of how you are ‘peers’ to the group participants. Some of this is hinted at as the piece develops, but it would help to include a clear statement of your positionality from the start.

Further details about the group are also needed: how often do you meet, and for how long? How large is the group? How long do participants typically remain involved? Setting the scene more clearly would help to contextualise the type of peer support group being discussed.

Another recommendation is that there remains space for further grounding in existing literature. The opening section is well referenced and draws upon appropriate global evidence in support of peer support models for mental health care. However, from this point forward all supporting literature is missing. For example, the description of the model of supervision of facilitators follows the established task-sharing model which could be referenced; equally in approaches to running the group, as well as in the facilitators lens section a number of recognised facilitation techniques are referred to – setting ground rules, boundary setting, demonstrating empathy, etc, all of which could be referenced. Equally, when discussing the therapeutic potential of peer support groups (page 4-5) there are claims made about safe spaces or the integration of arts-based activities, all of which could be supported by relevant literature.

Finally, the text is now quite long, and includes some sections and paragraphs that feel disjointed from the core focus. The section on reflecting on participants experiences strays from the facilitator lens, and repeats underpinning arguments for peer support which have already been made. It could be removed altogether. Equally lines 32-39 on page 5 discusses different people sharing experiences with the group, but I was struggling to see what this paragraph adds, and how it is relevant to ‘issues to consider’ which is the focus of this sub-section? Similarly for the paragraph about online groups, this doesn’t seem to relate to your experience, so I am not sure is relevant here? Please review the full piece to reduce words and repetition of points, and ensure clarity of focus on the facilitators’ perspective.

I hope these requests for minor revisions supports the authors in further refining their manuscript for publication.

---

## [Reviewer Report]

Many thanks to the authors for their engagement with the minor revisions suggested, and for clarifying their position on retaining the section on participants perspectives.

I enjoyed reading the revised manuscript, and am happy to support its publication. Please note, I did notice a few typos to address in the proofing process (e.g. p.2, line 52 reads “Having an allows....”, I assume there is a word missing here?).

I look forward to seeing this paper published. It offers a much needed and valued contribution to the global mental health literature. Thanks again to the authors for sharing their perspectives, experiences, and insights around peer support.